# Changes in Physicochemical Characteristics and Microbial Diversity of Traditional Fermented Vinasse Hairtail

Yue Zhang [1], Chuanhai Tu [1,*], Huimin Lin [1], Yuwei Hu [1], Junqi Jia [1], Shanshan Shui [1], Jiaxing Wang [2], Yi Hu [1] and Bin Zhang [1,3,*]

1   Zhejiang Provincial Key Laboratory of Health Risk Factors for Seafood, College of Food and Pharmacy, Zhejiang Ocean University, Zhoushan 316022, China
2   Research Office of Marine Biological Resources Utilization and Development, Zhejiang Marine Development Research Institute, Zhoushan 316021, China
3   Pisa Marine Graduate School, Zhejiang Ocean University, Zhoushan 316022, China
*   Correspondence: tu2021@zjou.edu.cn (C.T.); zhangbin@zjou.edu.cn (B.Z.); Tel.: +86-0580-255-4781 (C.T. & B.Z.)

**Abstract:** Fermented foods may confer several benefits to human health and play an important role in a healthy and balanced diet. Vinasse hairtail is a farmhouse-fermented food product with cultural and economic significance to locals in Zhoushan China. It is traditionally produced and subjected to 0–8 days of fermentation. In this study, we aimed to characterize the microbiota and physicochemical profiles of vinasse hairtail across different stages of fermentation. With the increase of fermentation time, pH, total sugar content, reducing sugar content, fat content, salt content, total protein content, myofibrillar protein content, TVB-N, and TBARS index increased, while the peroxide value decreased. The addition of vinasse significantly intensified the lipid and protein oxidation and protein degradation of hairtail, thereby increasing the flavor of its products. The microbial diversity and succession characterization during the fermentation of vinasse hairtail by high-throughput sequencing was measured. Results showed that Firmicutes was the predominant phylum and *Lactobacillus* was the main genera of bacterial diversity. Ascomycota was the main phylum of fungi and the main fungal genera detected in the samples were *Saccharomyces*. Additionally, the correlation between microbial community and physicochemical properties was found. Our study revealed that *Lactobacillus* was the major lactic acid bacteria present throughout the fermentation process. The results may provide a theoretical basis for improving the overall quality of vinasse hairtail.

**Keywords:** high-throughput sequencing; biodiversity; vinasse hairtail

## 1. Introduction

Traditional fermented food is one of the intangibles of human history [1]. Fish fermentation first emerged to preserve fresh fish, especially in hot and humid coastal areas [2]. The hairtail is one of the four famous economic fishes in the Zhoushan fishing ground (Zhejiang province, China). Nutritious and popular with consumers, the annual catch is more than 1 million tons. However, the hairtail has a short harvest period and is prone to spoilage, so vinasse hairtail as a means of storage was developed as a traditional folk wet-cured meat product. It is popular on the southeast coast of China, especially in Zhoushan city, Zhejiang province. Local consumers consider the vinasse hairtail as a specialty food product with unique flavor and texture. As the duration of the fermentation cycle increases, the physicochemical properties of the vinasse hairtail are altered, affecting the quality of the final products. The special flavor of vinasse hairtail is produced during fermentation by a series of complex biochemical reactions, including lipid oxidation and protein degradation. These changes mainly depend on enzymatic catalysis by endogenous and exogenous enzymes from microorganisms.

In recent years, the composition of microorganisms and the identification of dominant species in preserved aquatic products have been extensively studied. The microbiota plays a vital role in the production of volatiles in traditional fermented foods [3–6]. High-throughput sequencing (HTS) technology is an effective molecular tool for characterizing the diversity of microbial communities and has been successfully used to determine changes in microbial populations during the production or storage of many meat products [7–9].

As an example of mixed fermentation of plant and animal sources, the fermentation process of vinasse fish is worth further investigation. Little research has been done on microbial populations in traditional vinasse hairtail during production. Microorganisms play an important role in the formation of the flavor of the vinasse hairtail, and these microorganisms primarily come from the vinasse. As such, it is of great significance to study the microbial changes of vinasse. In the present study, we analyzed the physicochemical properties of vinasse hairtail at different stages of fermentation, and the composition and diversity of the microbial communities in the vinasse hairtail samples were investigated by high-throughput sequencing of bacterial 16S rRNA gene and fungal internal transcribed spacer (ITS1) gene amplicons. For studying the relationship between fermentation quality and microorganisms in vinasse hairtail, the differences in physicochemical characteristics and microbial diversity among the different stages of fermentation of vinasse hairtail have been detected, dispelling doubts in the minds of consumers. These results provide a reference basis for the development and production standardization of traditional processed products in the future.

## 2. Materials and Methods

### 2.1. Samples Collection

The preparation process of vinasse hairtail is shown in Figure 1. Fresh hairtails were purchased from a local supermarket in Zhoushan City, Zhejiang Province, China. After soaking at $20 \pm 5$ °C for 8 h, fresh glutinous rice was drained and steamed for 40 min, then cooled quickly to room temperature by rinsing it with cold sterile water. After draining, the rice was mixed with 1% ($w/w$) starter culture, sealed and fermented in an incubator at 30 °C for 2 days, then used as vinasse. The head and viscera of fresh hairtails were removed, and the remainder was cut into pieces then covered with salt at a ratio of 12% ($w/w$) before being pickled at 15 °C for 3 h. The excess salt on the surface of the hairtail pieces was washed away with running water, then the hairtail was dried in a drying oven at 55 °C with hot air until the moisture content of the fish reached about 55% ($w/w$), and stored in $-20$ °C for use. The dried hairtail and vinasse were evenly placed in a cleaned and dried fermenter according to the mass ratio of 1 to 2. A layer of vinasse was first placed at the bottom of the fermenter, and then the hairtail and vinasse were placed layer by layer. The top layer was sealed with vinasse, and fermentation was carried out in a 25 °C incubator.

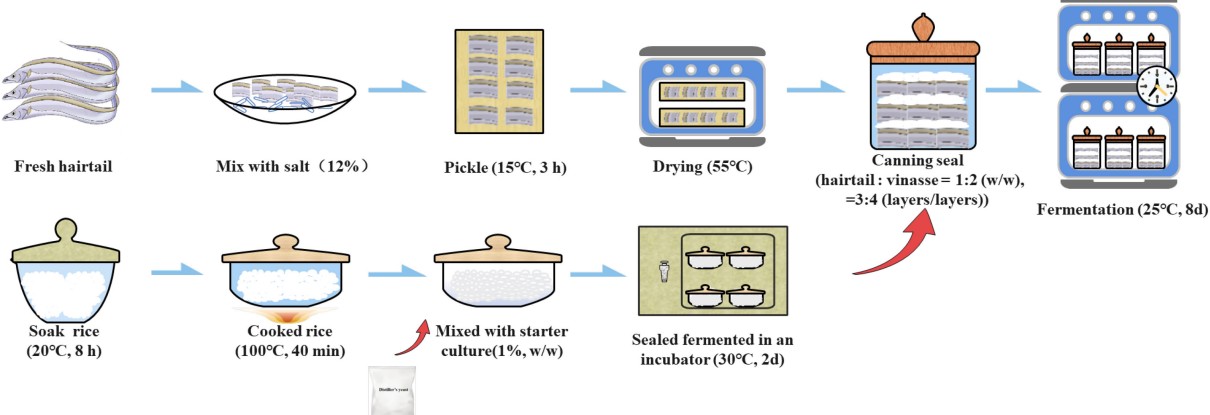

**Figure 1.** The schematic procedure of vinasse hairtail preparation.

## 2.2. Observation of Vinasse Hairtails with the Scanning Electron Microscope (SEM)

The morphology and microbial diversity of different samples were observed by using a scanning electron microscope (Hitachi Regulus 8100, Hitachi, Tokyo, Japan) at an acceleration voltage of 5 and 10 kV under magnifications of 2500×, 5000×, and 10,000×.

## 2.3. Texture Profile Analysis and Color Changes

The Lightness (*L*\*), green-red chromaticity (*a*\*) and green-red chromaticity (*b*\*) values of vinasse hairtail samples were measured by a CS-210 colorimeter (Hangzhou CHNSpec Technology Co., Ltd., Hangzhou, China). Each sample was measured three times on each side in three separate locations, with three replicates for each condition [10].

A Texture Analyser (Hangzhou, China) with a probe (P25) was used to compress (trigger force, 5g) a vinasse hairtail sample positioned on the center of the platform of the texture analyzer until it reached 50% deformation. The test speed, pre-test speed, and post-test speed were set as 1 mm/s, 1 mm/s, and 5 mm/s, respectively.

## 2.4. Physicochemical Determination

The fat content and total volatile basic nitrogen (TVB-N) were analyzed according to Chinese national standards using GB 5009.6,2016 [11] and GB 5009.228,2016 [12], respectively. The salt content was determined using a salinity meter (LH-Y28, Lohand Biological Co. Ltd., Hangzhou, China), and the pH value was determined using a digital pH meter (PHS-3C, Shanghai YiDian Scientific Instrument Co. Ltd., Shanghai, China). Total protein content, total sugar content, reducing sugar content, myofibrillar protein content, peroxide value, and 2-thiobarbituric acid reactive substance (TBARS) content were determined using reagent kits manufactured by Nanjing Jiancheng biological company.

## 2.5. DNA Extraction, PCR Amplification, and Sequencing

Samples were collected by suction filtration to filter the microorganisms in the vinasse hairtail liquid. Genomic DNA was extracted following the protocol of the FastDNA Spin Kit for Soil (MP Biomedicals, Norcross, GA, USA). The V3-V4 hypervariable region of the 16S rRNA gene was amplified using PCR with universal primer 338F (5′-ACTCCTACGGGAGGCAGCAG-3′) and 806R (5′-GGACTACHVGGGTWTCTAAT-3′) for analyzing the bacterial community. The ITS1 region was amplified using PCR with primer set ITS1F (5′-CTTGGTCATTTAGAGGAAGTAA-3′) and ITS2R (5′-GCTGCGTTCTTCATCGATGC-3′). The PCR reaction mixture included 4 μL 5 × FastPfu buffer, 2 μL 2.5 mM dNTPs, 0.8 μL each primer (5 μM), 0.4 μL FastPfu polymerase, 10 ng of template DNA, and ddH$_2$O to a final volume of 20 μL. PCR amplification cycling conditions were as follows: initial denaturation at 95 °C for 3 min, followed by 27 cycles of denaturing at 95 °C for 30 s, annealing at 55 °C for 30 s, and extension at 72 °C for 45 s, and single extension at 72 °C for 10 min, and finishing at 4 °C. All samples were amplified in triplicate. The PCR product was extracted from 2% agarose gel and purified using the AxyPrep DNA Gel Extraction Kit (Axygen Biosciences, Union City, CA, USA) according to the manufacturer's instructions and quantified using a Quantus™ Fluorometer (Promega, Madison, WI, USA).

Purified amplicons were pooled in equimolar amounts and paired-end sequenced on an Illumina MiSeq PE300 platform (Illumina, San Diego, CA, USA) according to the standard protocols by Majorbio Bio-Pharm Technology Co. Ltd. (Shanghai, China).

## 2.6. Data Processing and Statistical Analysis

All analyses were performed in triplicate. Physicochemical data were analyzed using the SPSS 22.0 statistical package (SPSS Inc. Chicago, IL, USA), and Pearson correlation analysis of bacteria/fungi and figures were done by Origin 2021. The experimental data were the mean values of three parallel experiments, and the results were expressed as mean ± standard deviation.

Raw FASTQ files were de-multiplexed using an in-house Perl script, then quality-filtered by fastp version 0.19.6 [13] and merged by FLASH version 1.2.7 [14] with the

following criteria: (i) the 300 bp reads were truncated at any site receiving an average quality score of <20 over a 50 bp sliding window, and the truncated reads shorter than 50 bp were discarded (reads containing ambiguous characters were also discarded); (ii) only overlapping sequences longer than 10 bp were assembled according to their overlapped sequence. The maximum mismatch ratio of the overlap region is 0.2. Reads that could not be assembled were discarded; (iii) Samples were distinguished according to the barcode and primers, and the sequence direction was adjusted to exact barcode matching. Then the optimized sequences were clustered into operational taxonomic units (OTUs) using UPARSE 7.1 [15,16] with a 97% sequence similarity level. The most abundant sequence for each OTU was selected as a representative sequence. To minimize the effects of sequencing depth on alpha and beta diversity measures, the number of 16S rRNA gene sequences from each sample was rarefied to 20,000, which still yielded an average Goods coverage of 99.09%, respectively.

The taxonomy of each OTU representative sequence was analyzed by RDP Classifier version 2.2 [17] with a confidence threshold of 0.7.

Bioinformatic analysis of the microbiota was carried out using the Majorbio cloud platform (https://cloud.majorbio.com, accessed on 1 December 2022). Alpha diversity was used to analyze species diversity for samples using 6 indices: Chao1 and ACE estimators were selected to measure community richness, Shannon and Simpson's indices were used to estimate community diversity, Good's coverage was used to characterize sequencing depth, and observed species was used to estimate the amount of unique OTUs found in each sample. These indices were calculated using QIIME and plotted using R (Version 2.15.3). Beta diversity was estimated for both weighted and unweighted UniFrac using QIIME to evaluate differences in species complexity across samples.

## 3. Results and Discussion

### 3.1. Macroscopic and Microscopic Properties of Vinasse Hairtail

As shown in Figure 2, fish pieces of vinasse hairtail from the early stage (0 d) and late stage (8 d) of fermentation featured different macro characteristics: fish pieces changed from white and slightly hard to yellow and soft during the fermentation.

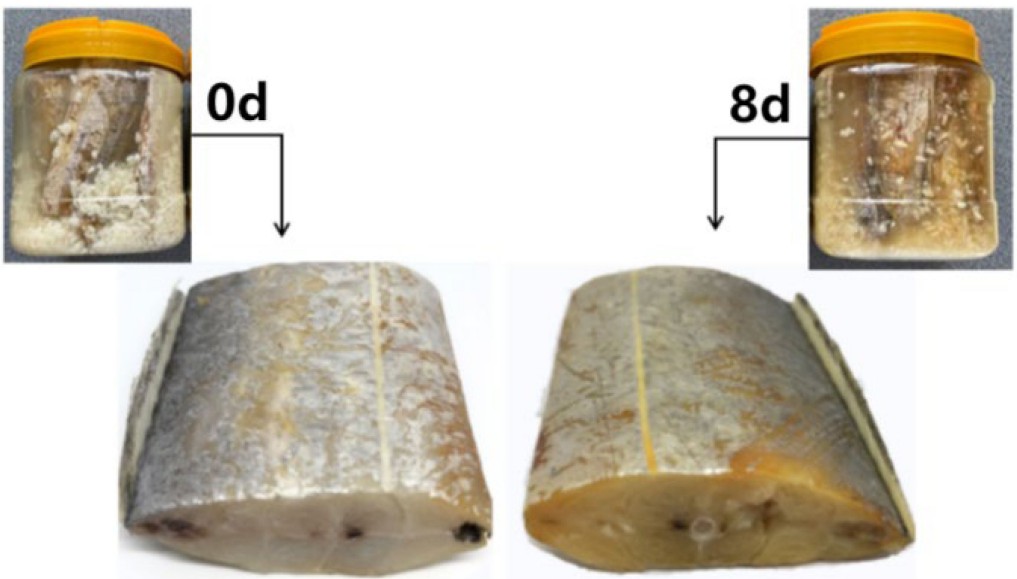

**Figure 2.** Macrostructure of vinasse hairtails muscle tissues with different fermentation.

Scanning electron microscopes of vinasse hairtails muscle tissues subject to different fermentation times demonstrated that fish rehydration, microbial growth and reproduction, and muscle tissue decomposition occurred in fermentation (Figure 3). At an early stage of fermentation (0 d), the folds of dehydrated muscle tissue were caused by the low water

content and activity of hairtail salted and air-dried with a certain concentration of salt. With increasing fermentation time, the myofibers began to swell. This was attributed to the solubilization/extraction of myofibrillar proteins and the swelling of filament lattice [18], presumably contributing to increased water-holding capacity. Concurrently, an observed debilitated connective tissue might affect the textural properties. The rupture or weakening of collagen cross-links, possibly by altering the stability of the helical structure of the molecule, might occur and affect the quality characteristics [19]. No traces of microorganisms can be observed from the vinasse hairtails muscle tissue in 0 d, but small traces of microorganisms can be seen in 4 d. The obvious and dense cell morphology of spheroidal and rod-shaped bacteria and yeast can be seen in 8 d. Results indicated that both bacteria and yeasts grew well during the fermentation process of the octopus, and the fermentation substrate of the vinasse hairtails was rich in nutrients and suitable for the growth and reproduction of microorganisms.

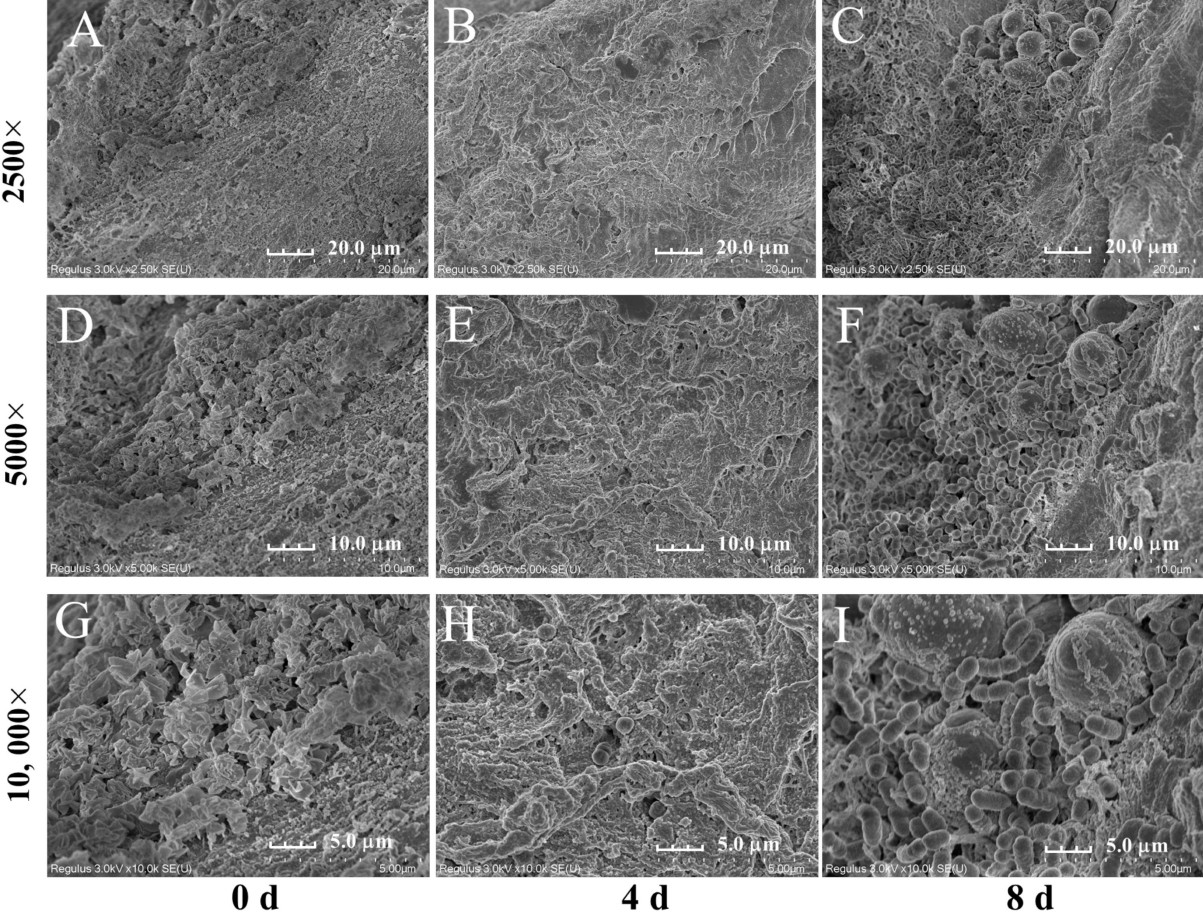

**Figure 3.** Scanning electron microscopes of vinasse hairtails muscle tissues with different fermentation times. At 2500× of the fermentation early stage (0 d) (**A**), the middle stage of fermentation (4 d) (**B**), and the late stage of fermentation (8 d) (**C**); at 5000× of 0 d (**D**), 4 d (**E**), and 8 d (**F**); at 10,000× of 0 d (**G**), 4 d (**H**), and 8 d (**I**).

### 3.2. Texture Profile Analysis and Color Changes

Color is the first factor that affects consumer acceptance of food products. The formation and stability of color are important for fish products. During the fermentation, the outside color value changes of *L** (lightness), *a** (redness), and *b** (yellowness) of fermented hairtails are presented in Table 1. Compared with the end of fermentation (8 d), at the beginning of fermentation (0 d), the color and brightness *L** value of hairtail showed a significant downward trend, while the redness *a** value and yellowness *b** value showed a significant

upward trend. High-temperature fermentation may accelerate the non-enzymatic browning reaction, such as oxidation of heme and fat, and the Maillard reaction, making it darker in color [20,21]. There was no significant difference between 4 d and 8 d, which may be because the intensity in the later stage of fermentation was significantly less than that in the first four days.

**Table 1.** Changes of texture and color parameters during the process of fermented vinasse hairtails.

|  | 0 d | 4 d | 8 d |
|---|---|---|---|
| Hardness (N) | 8903.77 ± 18.90 [c] | 5824.86 ± 30.25 [b] | 5110.09 ± 84.13 [a] |
| Sprigness (mm) | 0.76 ± 0.09 [a] | 0.79 ± 0.08 [a] | 0.82 ± 0.05 [a] |
| Cohesiveness | 0.88 ± 0.06 [a] | 0.72 ± 0.08 [a] | 0.58 ± 0.05 [a] |
| Chewiness (N) | 4994 ± 98.90 [b] | 2661.73 ± 112.79 [a] | 2577.47 ± 81.94 [a] |
| Shearing force (N) | 21.75 ± 0.91 [c] | 14.73 ± 1.58 [b] | 9.11 ± 0.51 [a] |
| $L^*$ | 60.65 ± 0.73 [a] | 50.58 ± 1.19 [b] | 48.18 ± 1.37 [b] |
| $a^*$ | −1.82 ± 1.18 [b] | 2.83 ± 1.05 [a] | 2.91 ± 0.67 [a] |
| $b^*$ | 10.14 ± 3.43 [b] | 17.54 ± 1.02 [a] | 18.89 ± 2.81 [a] |

Values are expressed at means ± SD ($n = 3$). Means with different lowercase letters in the same row are significantly different ($p < 0.05$).

Texture profile analysis (TPA) was used to reflect the texture characteristics of vinasse hairtail. After 8 days of fermentation, the hardness, chewiness, and shearing force of each sample decreased compared with 0 d, and there was no significant difference in springiness and cohesiveness. During the fermentation process, the increase of water content in hairtail meat and the more severe protein degradation can lead to the reduction of hardness, and at the same time help to improve the poor texture of hairtail meat caused by dehydration pretreatment before fermentation. The protease and lipase activity of microorganisms and the enzyme activity of the raw material degrade large molecules into small molecule compounds, which can lead to a looser texture and soft tissue of fermented vinasse hairtail [22].

### 3.3. Physicochemical Properties of Vinasse Hairtail at Different Stages of Fermentation

To investigate the product quality, the physicochemical properties of the hairtail at different stages of fermentation were examined. Physicochemical properties of the vinasse hairtail samples showed differences across the various fermentation times (Table 2). In food fermentation, the acidic environment could inhibit spoilage bacteria or pathogenic bacteria and ensure the normal growth of functional bacteria [23]. A low pH value is the endpoint characteristic of fermentation products. This improves the microbial stability of products by inhibiting pathogens [24]. During the vinasse hairtail fermentation process, the pH value is gradually decreased from 0 d (7.05 ± 0.03) to 8 d (4.54 ± 0.04). Compared to dry hairtail products, the addition of carbohydrates (glutinous rice vinasse) in the vinasse hairtail fermentation system can provide an extra energy source to accelerate the growth of microbes. Decomposition of carbohydrates often leads to a decrease in pH value and an accumulation of organic acids. The carbon source in the vinasse hairtail fermentation was primarily from the starch in glutinous rice, and results showed that total sugar, reducing sugar, fat, and total protein content decreased greatly with the fermentation.

The fat content results showed that the fat was broken down during fermentation. Lipid metabolism may have an important role in the formation of vinasse hairtail flavor, and lipolysis of the triglycerides and phospholipids by microbial and indigenous enzymes results in the development of free fatty acids. Lipolysis of phospholipids contributed significantly to the release of free fatty acids during fish fermentation. Furthermore, both microbial and endogenous lipases contributed to the liberation of free fatty acids in fermented fish while endogenous lipases also play a major role [25]. Peroxide value and TBARS levels are widely used as the primary and secondary indicators to evaluate lipid oxidation levels in fish and meat products [26,27]. As the fermentation proceeded, the peroxide value slightly increased from 0.07 ± 0.01 to 0.33 ± 0.01 g/100 g, while the TBARS

value ($p < 0.05$) decreased significantly from $7.67 \pm 0.03$ to $3.34 \pm 0.05$ nmol/mg, lower than 5 mg/kg, the maximum acceptable value for fish products [28]. Visessanguan et al. [29] reported a decreased TBARS value while the peroxide value increased during pork sausage fermentation. They suggested that only in oils containing three or more double bonds can TBARS have a positive correlation with peroxide value, and the reduction of TBARS value may be due to degradation of aldehydes or further reactions with amino acids to produce some flavor compounds.

**Table 2.** Physicochemical properties changes in vinasse hairtail during fermentation.

| | 0 d | 2 d | 4 d | 6 d | 8 d |
|---|---|---|---|---|---|
| pH | $7.05 \pm 0.03$ [e] | $6.22 \pm 0.01$ [d] | $5.64 \pm 0.05$ [c] | $5.22 \pm 0.02$ [b] | $4.54 \pm 0.04$ [a] |
| Total sugar content (g/L) | $36.06 \pm 1.14$ [d] | $25.81 \pm 0.60$ [c] | $16.49 \pm 0.63$ [b] | $15.92 \pm 0.46$ [b] | $13.81 \pm 0.22$ [a] |
| Reducing sugar content (g/L) | $26.06 \pm 0.59$ [e] | $18.96 \pm 0.83$ [d] | $10.02 \pm 0.01$ [c] | $8.66 \pm 0.13$ [b] | $7.59 \pm 0.12$ [a] |
| Fat content (%) | $9.21 \pm 0.18$ [e] | $7.66 \pm 0.11$ [d] | $6.17 \pm 0.04$ [c] | $5.34 \pm 0.14$ [b] | $4.72 \pm 0.16$ [a] |
| Salt content (%) | $11.10 \pm 0.30$ [e] | $10.03 \pm 0.01$ [d] | $9.77 \pm 0.02$ [c] | $9.37 \pm 0.05$ [b] | $8.95 \pm 0.05$ [a] |
| Total protein content (mg/g) | $161.66 \pm 0.47$ [d] | $128.11 \pm 0.89$ [c] | $86.47 \pm 1.02$ [b] | $74.83 \pm 0.84$ [a] | $74.65 \pm 0.35$ [a] |
| Myofibrillar protein content (mg/g) | $79.52 \pm 0.33$ [d] | $67.78 \pm 1.36$ [c] | $57.85 \pm 0.09$ [b] | $51.58 \pm 1.85$ [a] | $49.37 \pm 0.91$ [a] |
| TVB-N (mg/100g) | $38.65 \pm 0.38$ [e] | $26.96 \pm 0.51$ [d] | $20.61 \pm 0.50$ [c] | $15.78 \pm 0.74$ [b] | $14.40 \pm 0.30$ [a] |
| TBARS (nmol/mg) | $7.67 \pm 0.03$ [c] | $4.73 \pm 0.24$ [b] | $4.64 \pm 0.02$ [b] | $3.35 \pm 0.09$ [a] | $3.34 \pm 0.05$ [a] |
| Peroxide value (g/100 g) | $0.07 \pm 0.01$ [a] | $0.09 \pm 0.01$ [b] | $0.21 \pm 0.01$ [c] | $0.29 \pm 0.01$ [d] | $0.33 \pm 0.01$ [e] |

Values are expressed at means $\pm$ SD ($n = 3$). Means with different lowercase letters in the same row are significantly different ($p < 0.05$).

The salt content decreased rapidly due to the addition of wine lees rehydrating the 0-d hairtail meat (salted and air-dried treatment), which previously had a low moisture content and water activity. Salt prevents the growth of spoilage microorganisms due to its intrinsic antibacterial properties and it is conducive to the growth of halophilic and salt-tolerant microorganisms [30].

During the fermentation process, the proteins as substrates are broken down by the microbial community, and total protein content and myofibril protein content decreased gradually. Most researchers believe that the protein degradation of fermented meat products is the result of the combined action of endogenous proteases and microbial enzymes. Studies have shown that in the rapid fermentation process of silver carp, endogenous proteases could degrade both sarcoplasmic and myofibrillar proteins and *Lactobacillus plantarum* could hydrolyze sarcoplasmic peptides [31]. Endogenous proteases are mainly responsible for the hydrolysis of proteins into oligopeptides, while microbial enzymes contribute to the continuous degradation of oligopeptides into small peptides and free amino acids [32]. Similar conclusions were also found for fermented seafood [33,34]. Muscle endogenous proteases, including cysteine and lysosomal enzymes, catalyze the degradation of myofibrillar proteins at the initial phase of fish fermentation [35].

TVB-N could evaluate spoilage degrees from the protein levels, which is an important parameter in determining the shelf-life of fish [36,37]. TVB-N values of vinasse hairtail were decreased significantly ($p < 0.05$) by fermentation. This may be due to the spread of volatile nitrogen substances and malonaldehyde in hairtail to the surrounding vinasses after the fermentation process begins. Additionally, the reaction between acidic substances and alkaline nitrogenous substances during fermentation may also lead to a decrease in the TVB-N value. The TVB-N value after fermentation was 14.40 mg/100 g, much lower than the acceptable limit, $\leq 30$ mg/100 g, which is the safe edible standard defined by the European Commission [38,39].

*3.4. High-Throughput Metagenomic and Alpha Diversity Indexes Analysis of Vinasse Hairtail Microbial Community*

3.4.1. Alpha Diversity Indexes

Alpha diversity reflects the abundance and diversity of microbial communities and the alpha diversity index in each group of samples. As shown in Table 3, the sequencing

coverage of each group of samples was above 0.99, indicating that the sequencing had high coverage of most microorganisms in the samples, and the sequencing depth was suitable, thereby meeting the needs of diversity analysis (Alpha diversity) in each group of samples.

**Table 3.** The 16S rRNA and ITS diversity index of the vinasse hairtail samples at different stages of fermentation.

| | Samples | Sobs | Seq Num | OUT Num | Shannon Index | Simpson | ACE Index | Chao1 Index | Coverage |
|---|---|---|---|---|---|---|---|---|---|
| 16S rRNA | 0 d | 92 | 33,875 | 111 | 2.63 | 0.11 | 105.13 | 102.3 | 0.99 |
| | 4 d | 63 | 33,348 | 79 | 1.53 | 0.33 | 76.64 | 72.81 | 0.99 |
| | 8 d | 103 | 36,890 | 138 | 2.01 | 0.25 | 118.67 | 123.58 | 0.99 |
| ITS | 0 d | 31 | 56,393 | 36 | 1.18 | 0.41 | 35.24 | 35.28 | 0.99 |
| | 4 d | 19 | 38,140 | 28 | 0.24 | 0.91 | 31.55 | 27 | 0.99 |
| | 8 d | 30 | 45,877 | 44 | 0.72 | 0.57 | 32.57 | 32.56 | 0.99 |

Values are expressed at means ± SD (n = 3). Means with different lowercase letters in the same row are significantly different (*p* < 0.05).

Sobs represent the species number, and the Chao1 and ACE index are parameters of species richness. As shown by the results, the bacterial richness of the three groups constantly changed during the vinasse hairtail fermentation process. For the bacterial community, the ACE and Chao1 index in the 0 d and 8 d samples were all higher than those in 4 d, and 8 d was the highest one. These results indicate that the bacterial community richness decreased at 4 d and reached the highest at 8 d, while the fungal community richness was highest at 0 d, decreased at 4 d, and reached higher values at 8 d. This is probably because yeast is dominant in vinasse. The Shannon and Simpson, which are sampling diversity estimators, represent the approximate amounts of species and the evenness of their distribution [40]. Both bacterial and fungal communities have the same variation tendency: 0 d had higher bacterial diversity than 8 d, and 4 d had the lowest. Accordingly, it might be that the microorganisms were constantly adjusting and adapting to the relatively highly acidic fermentation environment from 0 d to 4 d, and various microorganisms eventually survived and grew at 8 d.

3.4.2. Bacterial and Fungal Community Dynamics during Vinasse Hairtail Fermentation

The OTU distribution of vinasse hairtail was also demonstrated by the Venn plot (Figure 4). Clustering was performed with OTUs defined at 97% sequence similarity. The Venn plot showed that 171 bacterial OTUs and 59 fungal OTUs were common across the three groups of vinasse hairtail samples. The highest OTUs were seen in the 8 d sample (138 bacteria and 44 fungi). It was consistent with the results of the above Alpha diversity indices analysis.

The dominant bacterial phylum in vinasse hairtail samples was significantly changed during fermentation (Table 4). In 0 d samples, Proteobacteria account for the highest proportion (82.81%), followed by Firmicutes (15.76%) and Actinobacteria (1.23%). For 4 d samples, Firmicutes rose to the highest proportion (95.36%) while Proteobacteria reduced (4.55%). The dominant microorganisms of dried salted fish largely included various bacteria, and yeasts are used as the main starter in rice wine [41,42]. Therefore, it was speculated that on the 0 d of fermentation, yeasts were primarily from rice wine while bacteria were mainly from dried hairtail. For 8 d samples, Firmicutes was the highest proportion (91.52%) and Proteobacteria was next (8.19%). At the family level, nine bacterial families (more than 1%) were observed during the fermentation. Lactobacillaceae belongs to the lactic acid bacteria, which can rapidly control the growth of the total bacterial colonies during fermentation and inhibited the growth of foodborne pathogenic bacteria [43]. This increased from 0.04% (0 d) to 54.17% (4 d), and eventually to 86.73% (8 d); Moraxellaceae decreased from 67.32% (0 d) to 1.94% (4 d), and eventually increased to 3.77% (8 d); Leuconostocaceae increased from 9.02% (0 d) to 40.02% (4 d), but eventually decreased to 0.46% (8 d); Enterobacteriaceae

decreased from 4.99% (0 d) to 2.30% (4 d), and eventually increased to 3.94% (8 d); and Staphylococcaceae decreased from 6.19% (0 d) to 0.13% (4 d), and eventually increased to 0.84%. Most of the Moraxinaceae belonging to Proteobacteria are anaerobic bacteria and cannot use carbohydrates to produce acid. As such, they decrease sharply in the anaerobic environment of the fermentation process. *Lactobacillus* belonging to Firmicutes mainly produce propionic acid, butyric acid, lactic acid, organic acid, and other amino acids that play an important role in product flavor and human health [44], while *Lactobacillus* and *Staphylococcus* of Firmicutes are the main bacteria in the fermented product and affect the flavor of salted fishes [45].

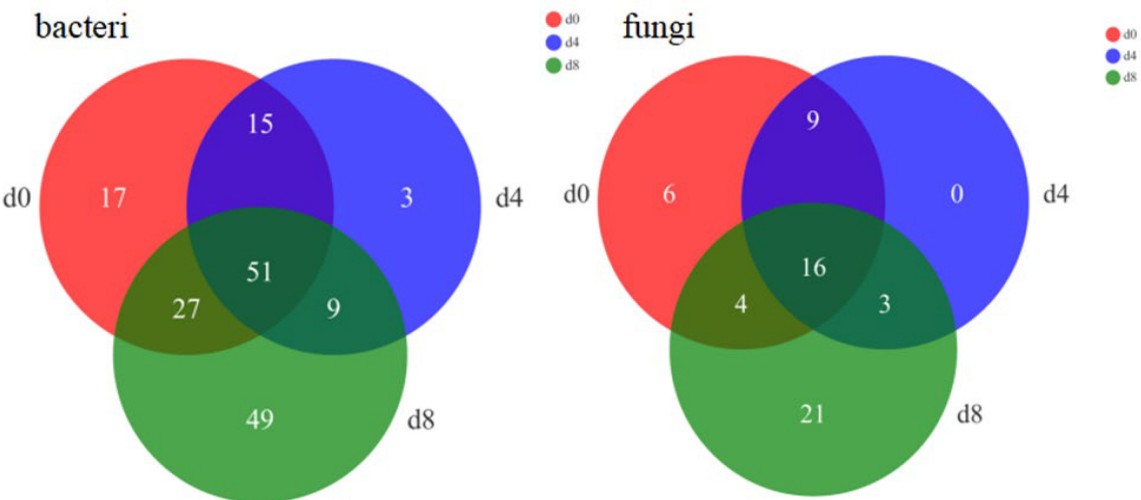

**Figure 4.** Venn analysis of OTU distribution of vinasse hairtail.

The fungal phylum dominantly presented in vinasse hairtail samples is Ascomycota and experienced no change during fermentation (Table 4). Ascomycota was the main group of fungi found in the vinasse hairtail samples. Ascomycota was the main group of fungi found in the Nuodeng ham samples and represented 98.4%, 95.6%, and 87.6% [46]. At the family level, four fungal families (more than 1%) were observed during the fermentation: Saccharomycetaceae (dominant microorganism), Phaffomycetaceae, and Metschnikowiaceae may come from rice vinasse, and Pichiaceae may come from hairtail pieces.

For a more detailed analysis of the bacterial dynamics associated with vinasse hairtail fermentation, bacterial compositions at the genus level were determined. At the genus level, 13 bacterial genera were detected during vinasse hairtail fermentation (Figure 5). The dominant bacterial genera before fermentation (0 d) were mostly *Psychrobacter* and *Acinetobacter*. These results are like those of studies on cured fish and ham [47,48]. Some studies have reported the ability of *Psychrobacter* sp. to produce histamine in a model cheese [49]. Similarly, some researchers have reported that *Enterobacter* was identified in fermented products, and it is proven that *Enterobacter* can produce large amounts of putrescine and cadaverine [50]. With the process of fermentation, *Lactobacillus* rapidly prolonged fermentation and was dominant at the end of the fermentation period, suggesting that the genus *Lactobacillus* probably played a key role in the fermentation of vinasse hairtail samples. Some studies have reported that the *Lactobacillus* population can affect metabolite formation during food fermentation, such as causing acidification of raw materials through the production of organic acids, primarily lactic acid [41,51]. *Lactobacillus* belongs to the lactic acid bacteria and plays an important role in food fermentation and preservation.

**Table 4.** Relative abundances of bacterial and fungal Phylum and Family levels (16 S) at different stages of fermentation.

| | | Relative Abundances (%) | | |
|---|---|---|---|---|
| **Bacteria** | **Phylum** | **0 d** | **4 d** | **8 d** |
| | Firmicutes | 15.76% | 95.36% | 91.52% |
| | Proteobacteria | 82.81% | 4.55% | 8.19% |
| | Actinobacteria | 1.23% | 0.07% | 0.23% |
| | others | 0.19% | 0.02% | 0.06% |
| | **Family** | | | |
| | Lactobacillaceae | 0.04% | 54.17% | 86.73% |
| | Enterobacteriaceae | 4.99% | 2.30% | 3.94% |
| | Moraxellaceae | 67.32% | 1.94% | 3.77% |
| | Vagococcaceae | 0.22% | 0.26% | 1.94% |
| | Staphylococcaceae | 6.19% | 0.13% | 0.84% |
| | Leuconostocaceae | 9.02% | 40.02% | 0.46% |
| | Halomonadaceae | 1.44% | 0.01% | 0.07% |
| | Pseudomonadaceae | 2.21% | 0.10% | 0.05% |
| | Acetobacteraceae | 6.49% | 0.06% | 0.00% |
| | Others | 2.08% | 1.01% | 2.20% |
| **Fungi** | **Phylum** | **0 d** | **4 d** | **8 d** |
| | Ascomycota | 99.90% | 99.97% | 99.78% |
| | others | 0.10% | 0.03% | 0.22% |
| | **Family** | | | |
| | Saccharomycetaceae | 55.85% | 95.08% | 70.64% |
| | Phaffomycetaceae | 29.59% | 3.14% | 27.37% |
| | Pichiaceae | 0.00% | 0.00% | 1.50% |
| | Metschnikowiaceae | 14.18% | 1.73% | 0.04% |
| | Others | 0.38% | 0.05% | 0.45% |

At the genus level, the relative abundance of different fungi microorganisms in all samples at the phylum level are shown in Figure 5. Five fungal genera were identified according to the sequences obtained from all samples: *Saccharomyces*, *Wickerhamomyces*, unclassified_f_Metschnikowiaceae, *Clavispora*, and *Pichia*. Vinasse hairtail samples were dominated by *Saccharomyces* and *Wickerhamomyces*, which are thought to contribute to rice wine production by providing good flavor and quality [42,52].

The correlation between physicochemical properties and microorganism genera in vinasse hairtail during different fermentation times was shown in Figure 6. The results indicated that *Lactobacillus*, *Acinetobacter*, unclassified_f_Enterobacteriaceae, and *Gluconobacter* in bacterial genus correlate with physicochemical properties. The *Lactobacillus* genus played an important role in the change of physicochemical properties and was negatively correlated with pH, total sugar content, reducing sugar content, fat content, salt content, total protein content, myofibrillar protein content, TVB-N, and TBARS, but positively correlated with peroxide value. The *Acinetobacter* and *Gluconobacter* genera were found to be negatively associated with peroxide value and significantly ($p < 0.05$) negatively correlated with pH, total sugar content, reducing sugar content, fat content, salt content, total protein content, myofibrillar protein content, TVB-N, and TBARS. The unclassified_f_Enterobacteriaceae was negatively correlated with peroxide value and positively correlated with pH, total sugar content, fat content, salt content, total protein content, Myofibrillar protein content, and TVB-N. The unclassified_f_Metschnikowiaceae and the *Clavispora* genus in the fungal genera significantly ($p < 0.05$) affected the physicochemical properties of vinasse hairtail, was found to be negatively correlated with peroxide value, and significantly ($p < 0.05$) positively correlated with pH, total sugar content, reducing sugar content, fat content, salt content, total protein content, myofibrillar protein content, TVB-N, and TBARS. In addition, the *Saccharomyces* genus was significantly ($p < 0.05$) negatively correlated with total sugar content, reducing sugar content, and total protein content. In general, bacteria influenced

the physicochemical properties more closely than fungus in the fermentation process of vinasse hairtail.

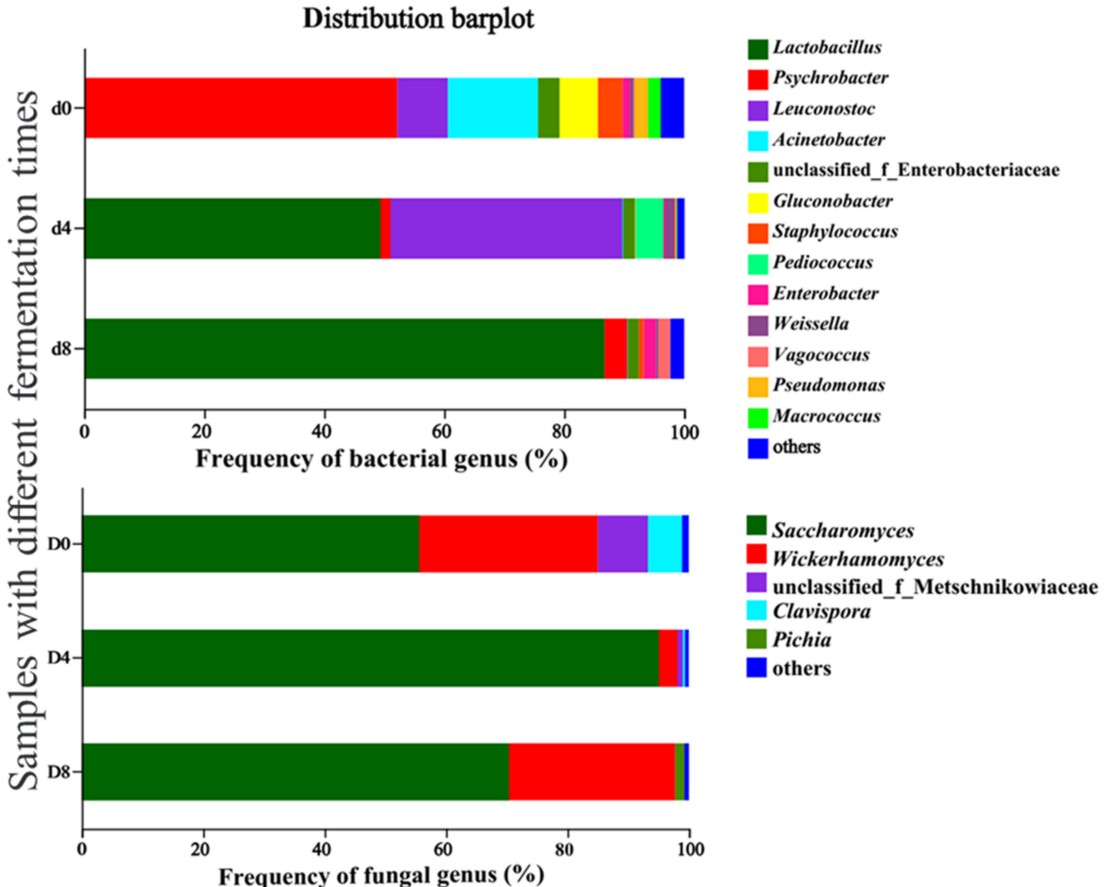

**Figure 5.** Barplots of bacterial and fungal diversity of vinasse hairtail at the genus level. Others: The sum of all genera that frequency less than 1%, unclassified_f = unclassified genus of a Family.

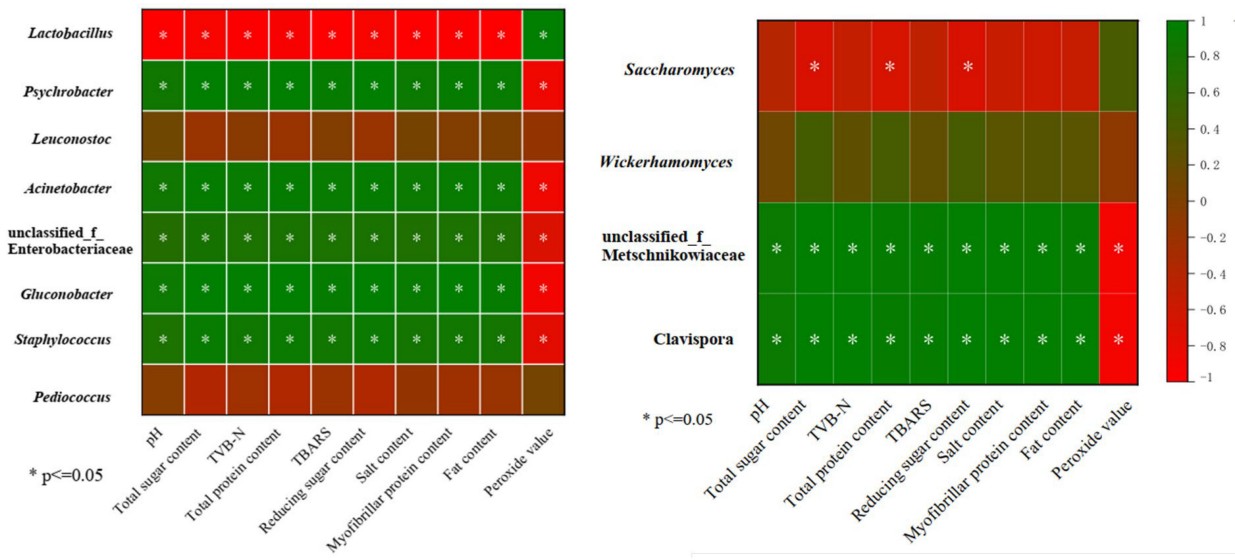

**Figure 6.** Heatmap of the correlation between physicochemical properties and microorganism genera in vinasse hairtail.

## 4. Conclusions

In this study, the physicochemical characteristics and microbial diversity of traditional fermented vinasse hairtail was evaluated. Results showed that vinasse and hairtails provided sufficient nutrients for the growth of microorganisms. The addition of vinasse as a starter culture and fermentation carbon source intensified the lipid and protein oxidation and protein degradation of hairtail, thereby increasing the flavor of its products. The composition of microbial communities varied at different stages of fermentation and affected the quality of the vinasse hairtail. *Lactobacillus* was the main genera of bacterial diversity at the end of the fermentation process, and the main fungal genera were *Saccharomyces*. The change of physicochemical properties was related to a variety of microorganisms, which together affect the quality of vinasse hairtail. These results may provide some preliminary information on a theoretical basis for improving the overall quality of vinasse hairtail.

**Author Contributions:** Conceptualization, B.Z. and C.T.; writing—original draft preparation, Y.Z., Y.H. (Yuwei Hu) and J.J.; writing—review and editing, C.T., S.S., H.L., Y.H. (Yi Hu) and J.W.; project administration, B.Z. and C.T.; funding acquisition, C.T. and J.W. All authors have read and agreed to the published version of the manuscript.

**Funding:** This work was supported by the National Natural Science Foundation of China (NSFC) (No. 32202187); the Zhoushan Science and Technology Project (Grant No. 2022C41025); Zhejiang Province Public Welfare project (Grant No. LGN21C190001).

**Data Availability Statement:** Data is contained within the article.

**Conflicts of Interest:** The authors declare no conflict of interest.

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
