# Peer review of "Changes in Physicochemical Characteristics and Microbial Diversity of Traditional Fermented Vinasse Hairtail"

_fermentation, doi:10.3390/fermentation9020173_

Round 1

Reviewer 1 Report

The study is interesting. However, all data of control are missed that make great default of this paper for comparison of the current data without vinasse and without LAB addition. Especially the authors claimed that The addition of vinasse intensified the lipid and protein oxidation and protein degradation of hairtail, thereby increasing the flavor of its products.

The authors should add these data

Besides some correction should be done

1-keyword delete (fermentation)

L43 correct Themicrobiota

The authors should add more data for the amount of hairtails/importance/problems and usual consumption

Please add space before reference no—see for example L31-46-88-95-…..etc

Rephrase sentence at L58-61 and clarify the aim of this study

Fig1—add more clarifications to be more clear…for example the layers of rice and hairtail –rations –temperature…etc

L111- correct ddH2O

Fig3 show the magnifications bars

L197 correct Textureprofile

Please show the data for control samples to clarify the real changes

L318 correct Staphylococcus

Table 4 and 5 please delete a and b and show each as one Table

Fig7 correct bacteri//correct X and Y axes titles and clarify the abbreviations

Author Response

Dear reviewer,

Thank you for the comments on our manuscript titled "Changes in physicochemical characteristics and microbial diversity of traditional fermented vinasse hairtail" (No. fermentation-2176369).

These comments provided important guidance for our following research. We addressed the reviewers’ comments to the best of our abilities and revised the text to meet the requirements. We hope this meets your requirements for a publication. We marked the revisions in red in the manuscript. The main comments and specific responses are detailed below:

The study is interesting. However, all data of control are missed that make great default of this paper for comparison of the current data without vinasse and without LAB addition. Especially the authors claimed that The addition of vinasse intensified the lipid and protein oxidation and protein degradation of hairtail, thereby increasing the flavor of its products.

The authors should add these data

Response: Thank you for your kind comments and suggestions. The existing research has shown that the dominant microorganisms of dried salted fish largely included various bacteria, and yeasts are used as the main starter in rice wine, so the fungus and bacteria in the 0d sample could roughly reflect the microbial composition of dried fish and rice wine, and considering that the focus of this study is vinasse hairtail and its fermentation process, we mostly compared the data before and after fermentation, and as per your suggestion, we have cited some related references about vinasse and dried fish in the manuscript.

Besides some correction should be done

1-keyword delete (fermentation)

Response: keyword “fermentation” has been deleted.

L43 correct Themicrobiota

Response: Themicrobiota now reads “The microbiota”.

The authors should add more data for the amount of hairtails/importance/problems and usual consumption

Response: Thank you for your suggestion, we have added relevant contents to the Introduction part.

Please add space before reference no—see for example L31-46-88-95-…..etc

Response: Space has been added before each reference no.

Rephrase sentence at L58-61 and clarify the aim of this study

Response: This sentence has been rephrased and now reads “For studying the relationship between fermentation quality and microorganism in vinasse hairtail, dispelling doubts in the minds of consumers, the differences in physicochemical characteristics and microbial diversity among the different stages of fermentation of vinasse hairtail have been detected, these results provided a reference basis for the development and production standardization of traditional processed products in the future.”

Fig1—add more clarifications to be more clear…for example the layers of rice and hairtail–rations –temperature…etc

Response: Fig1 has been remade with clarifications including the layers of rice and hairtail, rations, and temperature.

L111- correct ddH2O

Response: It has been corrected as “ddH2O”.

Fig3 show the magnifications bars

Response: The magnifications and plotting scales have been added in Fig3.

L197 correct Textureprofile

Response: It has been corrected as “Texture profile”.

Please show the data for control samples to clarify the real changes

Response: The existing research has shown that the dominant microorganisms of dried salted fish largely included various bacteria, and yeasts are used as the main starter in rice wine, so the fungus and bacteria in the 0d sample could roughly reflect the microbial composition of dried fish and rice wine, and considering that the focus of this study is vinasse hairtail and its fermentation process, we mostly compared the data before and after fermentation, and as per your suggestion, we have cited some related references about vinasse and dried hairtail fish in the manuscript.

L318 correct Staphylococcus

Response: It has been corrected as “Staphylococcus”.

Table 4 and 5 please delete a and b and show each as one Table

Response: Tables 4 and 5 have been changed and contents are merged into a new Table 4.

Fig7 correct bacteri//correct X and Y axes titles and clarify the abbreviations

Response: Fig 7 has been remade and the abbreviations were clarified in the figurehead.

Thank you again for your constructive advice and hope to meet your criterion. We don’t know whether it has reached the standard. If there are still some mistakes in the manuscript, please give us another chance to improve the manuscript. We will further revise the sentences according to the comments.

Sincerely,

Tu Chuanhai (Corresponding author)

Reviewer 2 Report

The manuscript by Zhang et al. describes the physicochemical changes during the fermentation process vinasse hairtail, as well as changes in the composition of the microbial community (bacteria and fungi). In general, the results obtained are undoubtedly of interest primarily to manufacturers of such traditional food products. Understanding the changes in microbial communities in the process of obtaining fermentable products makes it possible to develop standard approaches for their preparation. However, there are a number of comments about the manuscript:

1. When studying the composition of microbial communities, no biological replicas were made. This does not allow assessing the statistical significance of the data obtained. For example, judging by the obtained results of taxonomic diversity, during the fermentation process, the succession of microbial communities occurred and the number of dominant phyla and families decreased by the 8th day. However, according to the data obtained, the diversity indices increased, which looks strange. And on day 4, the dominant Leuconostocaceae family (40%), although its share is significantly lower on days 0 and 8. This is rather strange and perhaps this is just a deviation associated with a particular sample, and not with the community as a whole. So replicas are needed.

2. It is not entirely clear why physicochemical parameters in vinasse hairtail were sampled on days 0,2,4,6, and 8, while taxonomic diversity was assessed only on days 0, 4, and 8. To identify the correlation between different physicochemical parameters and the composition of microorganisms, it is worth taking the same points and conducting a PCA analysis.

3. In the analysis of microbial communities, control points are also missing - this is (1) the composition of vines microorganisms before adding (2) the composition of the starter culture that was added to cooked rice. (3) composition of swabs from dried hairtails. Comparison of these points will allow us to trace the assessment of the introduced groups, as well as the role of starter cultures in subsequent fermentation.

4. It is good to evaluate the proportion of bacteria and fungi at different stages of fermentation.

Author Response

Dear reviewer,

Thank you for the comments on our manuscript titled "Changes in physicochemical characteristics and microbial diversity of traditional fermented vinasse hairtail" (No. fermentation-2176369).

These comments provided important guidance for our following research. We addressed the reviewers’ comments to the best of our abilities and revised the text to meet the requirements. We hope this meets your requirements for a publication. We marked the revisions in red in the manuscript. The main comments and specific responses are detailed below:

The manuscript by Zhang et al. describes the physicochemical changes during the fermentation process vinasse hairtail, as well as changes in the composition of the microbial community (bacteria and fungi). In general, the results obtained are undoubtedly of interest primarily to manufacturers of such traditional food products. Understanding the changes in microbial communities in the process of obtaining fermentable products makes it possible to develop standard approaches for their preparation. However, there are a number of comments about the manuscript:

1. When studying the composition of microbial communities, no biological replicas were made. This does not allow assessing the statistical significance of the data obtained. For example, judging by the obtained results of taxonomic diversity, during the fermentation process, the succession of microbial communities occurred and the number of dominant phyla and families decreased by the 8th day. However, according to the data obtained, the diversity indices increased, which looks strange. And on day 4, the dominant Leuconostocaceae family (40%), although its share is significantly lower on days 0 and 8. This is rather strange and perhaps this is just a deviation associated with a particular sample, and not with the community as a whole. So replicas are needed.

Response: we tested the composition of microbial communities in triplicate, and results recorded were means, considering that standard deviations of the composition of microbial communities are generally not shown in current research papers, and too small standard deviations in experimental results are all statistically significant, so standard deviations are not shown in the results.

Diversity is an indicator to evaluate the microbial community of samples, focusing on different microbial types, which do not correlate with the number of dominant phyla and families. If the number of dominant phyla and families decreases, while new phyla or families are appearing in the test results of the sample, it means that the diversity index will increase.

As for the changes in the Leuconostocaceae family, we thought it was due to the succession of microbial communities, like the Lactobacillaceae family, from 0.04% (0d) to 54.17% (4d), and 86.73% (8d). And according to current research, the decrease in pH value and increase in the Lactobacillaceae family may result in a decrease in other bacteria like the Leuconostocaceae family. The relevant expressions have been added to the manuscript.

2. It is not entirely clear why physicochemical parameters in vinasse hairtail were sampled on days 0,2,4,6, and 8, while taxonomic diversity was assessed only on days 0, 4, and 8. To identify the correlation between different physicochemical parameters and the composition of microorganisms, it is worth taking the same points and conducting a PCA analysis.

Response: The purpose of sampling physical and chemical parameters every two days is to comprehensively evaluate the indexes of local traditional fermentation samples and determine the end point of fermentation. Samples 0, 4, and 8d for taxonomic diversity evaluation were taken to compare the microbial changes before and after fermentation, particularly identifying the microbial types in the final product.

As per your suggestions, the PCA analysis was conducted and shown in Fig. 8.

3. In the analysis of microbial communities, control points are also missing - this is (1) the composition of vines microorganisms before adding (2) the composition of the starter culture that was added to cooked rice. (3) composition of swabs from dried hairtails. Comparison of these points will allow us to trace the assessment of the introduced groups, as well as the role of starter cultures in subsequent fermentation.

Response: The composition of the starter culture that was added to cooked rice was mostly yeast, including Saccharomyces cerevisiae and Wickerhamomyces anomalus. The existing research has shown that the dominant microorganisms of dried salted fish largely included various bacteria, and yeasts are used as the main starter in rice wine, so the fungus and bacteria in the 0d sample could roughly reflect the microbial composition of dried fish and rice wine. Considering that the focus of this study is vinasse hairtail and its fermentation process, we mostly compared the data before and after fermentation, and as per your suggestion, we have cited some related references about vinasse and dried fish in the manuscript.

4. It is good to evaluate the proportion of bacteria and fungi at different stages of fermentation.

Response: The authors agree with the reviewer’s concerns about the evaluation of the proportion of bacteria and fungi at different stages of fermentation. The plate counting analysis of bacteria (including lactic acid bacteria, Enterobacteriaceae, and total bacterial count) and fungi in vinasse hairtail during fermentation is still in progress. The current manuscript, as we can see, has included 8 figures and 4 tables, so it’s not easy to add more results to the text. In the case of microbiological analysis, another new manuscript will be prepared and submitted to clarify the proportion of bacteria and fungi.

Thank you again for your constructive advice and hope to meet your criterion. We don’t know whether it has reached the standard. If there are still some mistakes in the manuscript, please give us another chance to improve the manuscript. We will further revise the sentences according to the comments.

Sincerely,

Tu Chuanhai (Corresponding author)